# Anchored Alignment for Self-Explanations Enhancement

## Abstract

In this work, we introduce a methodology for alignment designed to enhance the ability of large language models (LLMs) to articulate their reasoning—*self-explanation*—even in the absence of annotated rationale explanations. Our alignment methodology comprises three key components: explanation quality assessment, self-preference dataset generation, and model alignment. Additionally, we present a novel technique called *Alignment with Anchor Preference Pairs*, which improves the selection of preference pairs by categorizing model outputs into three groups: consistently correct, consistently incorrect, and variable. By applying tailored strategies to each category, we enhance the effectiveness of Direct Preference Optimization (DPO). Our experimental results demonstrate that this approach significantly improves explanation quality while maintaining accuracy compared to other fine-tuning strategies.

## 1 Introduction

Large language models (LLMs) have demonstrated remarkable capabilities across various tasks. However, fine-tuning these models for specific applications often leads to a critical trade-off: improvements in one area may compromise the model's generalization capabilities (Yang et al., 2024; Kirk et al., 2024). In our study, we aim to enhance a secondary task—specifically, the model's ability to articulate reasoning processes in natural language, a skill known as *self-explanation* (Madsen et al., 2024a)—in parallel with the primary task, despite the constraint of not having human-annotated rationales.

The lack of annotated data of both high- and low-quality explanations can be framed in the context of model aligning without human preference data. Recent research has explored ways to align LLMs without direct human input. Some approaches generate self-instruct data to fine-tune models (Wang et al., 2023; Chen et al., 2023; Gulcehre et al., 2023), while others, like Bai et al. (2022); Yuan et al. (2024); Wu et al. (2024), use LLM-generated feedback to train reward models. Building on these advancements, we propose an end-to-end approach to align LLMs on classification tasks while also ensuring the generation of high-quality self-explanations, even without annotated data for this secondary task. Our approach integrates three core components: evaluating generated explanations, creating self-preference datasets, and aligning the model. Additionally, we introduce *Alignment with Anchor Preference Pairs*, a method that improves preference pair selection by categorizing model responses into three groups: consistently correct, consistently incorrect, and variable. For each category, we apply tailored strategies to construct preference pairs, which are then used in the Direct Preference Optimization (DPO) phase (Rafailov et al., 2023). Our results demonstrate that this method consistently improves explanation quality, mitigating the degradation caused by SFT. Moreover, we show that using anchor preference pairs outperforms self-alignment strategies that rely solely on judge-based evaluations for preference pair selection.

Our contributions are summarized as follows:

*(i).* We introduce a framework for the qualitative assessment of self-explanations, designed to evaluate how effectively the model conveys its reasoning.

*(ii).* We analyze how supervised fine-tuning for classification tasks affects the quality of self-explanations. Our findings demonstrate that while SFT improves classification accuracy, it often reduces explanation quality, underscoring the need for improved alignment strategies.

*(iii).* We propose a novel method, *Alignment with Anchor Preference Pairs*, for constructing high-quality preference pairs when building self-preference datasets. This method uses the model's behavior on each input prompt to apply specific strategies while creating preference pairs. Our approach consistently outperforms other methods that rely solely on judge-based evaluations for selecting preference pairs.

*(iv).* We develop an end-to-end methodology for aligning LLMs to downstream classification tasks while maintaining the quality of their self-explanations, even in the absence of explanation-rich datasets.

## 2 A FRAMEWORK FOR QUALITATIVE ASSESSMENT OF SELF-EXPLANATIONS

### 2.1 QUALITY CRITERIA FOR EFFECTIVE SELF-EXPLANATIONS

We focus on the model's ability to provide a holistic explanation, which we define as one that excels across multiple criteria, collectively determining what constitutes a good explanation. This approach contrasts with previous work that emphasized specific trustworthiness metrics, such as faithfulness (Madsen et al., 2024b;a; Lanham et al., 2023; Lyu et al., 2023; Turpin et al., 2023; Parcalabescu & Frank, 2024) and truthfulness (Zhang et al., 2024; Sharma et al., 2023; Burns et al., 2022; Joshi et al., 2024). A high-quality, holistic explanation may be unfaithful in the sense that it does not accurately represent the model's internal reasoning, or conversely, a faithful explanation that truly reflects the model's reasoning may suffer from a lack of clarity, reducing its quality. While aiming for explanations that fulfill a holistic explanation framework might incidentally enhance faithfulness, it is not a strict requirement.

We evaluate self-explanations based on the following criteria:

1. **Logical coherence**: The explanation should follow a clear and logical reasoning process, with all components cohesively connected to form a unified, non-contradictory narrative.

2. **Clarity**: The explanation must present ideas clearly and precisely, using appropriate terminology to effectively communicate complex concepts without unnecessary complexity.

3. **Relevance**: The explanation should comprehensively address the task at hand, directly answering the specific context or requirements without omitting critical information.

4. **Depth of argumentation**: The explanation must provide strong reasoning and credible evidence to support its conclusions, reflecting a deep understanding of the task.

5. **Factual accuracy**: This criterion assesses the correctness of individual claims within the explanation. While related to truthfulness, factual accuracy focuses on whether specific statements align with established knowledge.

### 2.2 SELF-EXPLANATIONS EVALUATION METHODOLOGY

Let $\mathcal{M}$ represent a large language model tasked with generating responses for a classification problem. Each response consists of two components: a self-explanation, denoted as $\varepsilon_i$, and a predicted classification label, $\hat{y}_i$, corresponding to an input prompt $x_i$. The self-explanation $\varepsilon_i$ is produced by prompting the model to articulate its reasoning before providing a final prediction, following the Chain-of-Thought prompting strategy (Wei et al., 2022).

Our methodology is inspired by recent approaches that utilize LLMs as evaluators of other models' outputs (Dubois et al., 2023; Li et al., 2024; Fernandes et al., 2023; Bai et al., 2023; Saha et al., 2024). This approach has shown versatility, extending beyond simple evaluation to various applications in model improvement and self-alignment strategies. For instance, researchers have employed this framework to generate self-instruct data for fine-tuning models (Wang et al., 2023; Chen et al., 2023; Gulcehre et al., 2023) and to create feedback for training reward models (Bai et al., 2022; Yuan et al., 2024; Wu et al., 2024).

For our evaluation, we employ a more capable model, $\mathcal{M}_{\text{Judge}}$, to assess the quality of self-explanations $\varepsilon_i$ based on predefined criteria (detailed in Section 2.1). The evaluation process proceeds as follows:

1. For each criterion $\kappa$, $\mathcal{M}_{\text{Judge}}$ assigns a qualitative verdict $v_{i,\kappa}$ from the set {excellent, good, fair, poor, bad}. The prompt used by $\mathcal{M}_{\text{Judge}}$ is provided in Appendix F.2.

2. Each verdict $v_{i,\kappa}$ is mapped to a numerical score $s_{i,\kappa}$ (see Appendix F.1).

3. The overall score for an explanation, $s_i$, is computed as the sum of scores across all criteria:
$s_i = \sum_{k=1}^{K} s_{i,k}$

To assess the quality of self-explanations generated by different models, we adopt a pairwise evaluation (see Appendix A) strategy consistent with previous work (Chen et al., 2023; Yuan et al., 2024; Wu et al., 2024).

## 3 SELF-EXPLANATION ALIGNMENT WITH ANCHOR PREFERENCE PAIRS

In this section, we introduce a methodology for alignment designed to enhance the ability of large language models (LLMs) to articulate their reasoning—*self-explanation*—even in the absence of annotated rationale explanations. However, we assume access to human-annotated data in the form of classification datasets for domain-specific adaptation, reflecting a common constraint in real-world applications, where comprehensive explanation data is often scarce or prohibitively expensive compared to classification datasets.

Building on prior work (Bai et al., 2022; Wang et al., 2023; Yuan et al., 2024; Wu et al., 2024), our alignment methodology incorporates familiar components such as self-preference dataset generation, human-free evaluation of candidate responses using *LLM-as-Judge*, preference pair selection, and model alignment.

However, our approach differs from previous methods in two key ways: First, for the assessment of candidate responses, we use the evaluation explanation quality framework introduced in Sections 2.1 and 2.2. Second, we propose a novel technique, *Alignment with Anchor Preference Pairs*, which improves preference pair selection by categorizing model outputs into three groups: consistently correct, consistently incorrect, and variable. By applying tailored strategies to each category, we enhance the effectiveness of DPO.

The steps of the methodology are as follows:

1. Supervised fine-tuning of the base model $\mathcal{M}_{\text{Base}}$ specifically on a target classification task, resulting in $\mathcal{M}_{\text{SFT}}$.

2. Instruct $\mathcal{M}_{\text{SFT}}$ to generate multiple explanation-prediction pairs for each prompt, and evaluate the quality of these self-explanations using the methodology outlined in Sections 2.1 and 2.2. During alignment, the base model $\mathcal{M}_{\text{Base}}$ acts as the judge $\mathcal{M}_{\text{Judge}}$, ensuring the process remains self-contained.

3. Construct an alignment dataset by selecting preference pairs using an anchor-based strategy (see Section 3.3).

4. Align $\mathcal{M}_{\text{SFT}}$ via DPO with the dataset created in the third step, producing the aligned model $\mathcal{M}_{\text{Anchor}}$.

### 3.1 SUPERVISED FINE-TUNING WITHOUT ANNOTATED EXPLANATIONS

We fine-tuned the base model, $\mathcal{M}_{\text{Base}}$, on classification datasets (the primary task) to obtain $\mathcal{M}_{\text{SFT}}$, simulating scenarios where explanation annotations are unavailable. To replicate typical domain-specific adaptations and avoid potential gains from multi-task learning, we fine-tuned a separate model for each task. During fine-tuning, loss was calculated only on the target tokens corresponding to the correct choice sentence, excluding the system instruction and question. We generated the full text of the selected option to provide richer context and preserve the model's text generation capabilities. Details on datasets and training setups are provided in Section 4.1.

### 3.2 SELF-PREFERENCE DATASET CREATION

We generate self-preference data for alignment as follows:

1. **Generate candidate responses:** We sample $N$ diverse pairs of explanations and predictions from $\mathcal{M}_{\text{SFT}}$, denoted as $\{\varepsilon_i^n, \hat{y}_i^n\}_{n=1}^N$, where $\varepsilon_i^n$ represents the explanation for the $n$-th prediction $\hat{y}_i^n$ corresponding to the prompt $x_i$.

2. **Evaluate candidate responses:** We use the methodology described in Section 2.2 to evaluate the self-explanations generated from the candidate responses, assigning a score $s_i^n$ to each explanation $\varepsilon_i^n$. During the creation of the self-preference dataset, we employ $\mathcal{M}_{\text{base}}$ as the judge ($\mathcal{M}_{\text{Judge}}$). This ensures that the model alignment process remains self-contained, without the need for external models, except for evaluation purposes. The decision to use $\mathcal{M}_{\text{base}}$ as the judge stems from the observation that $\mathcal{M}_{\text{STF}}$ exhibits a decline in explanation quality (see Section 4.1.1) and reduced agreement with the evaluation judge (see Appendix C).

### 3.3 Preference Pairs via Anchor Selection

We introduce a method to enhance the selection of preference pairs by categorizing model responses into three groups: consistently correct, consistently incorrect, and variable. For each category, we apply specific strategies to construct preference pairs, which are then used in during the DPO phase. To evaluate the model's consistency on a given input prompt, a ground truth reference, or *anchor*, is required. We use a classification task as the probing mechanism.

**Preference Pairs for Consistently Correct Prompts**: For input prompts $x_i$ where $\mathcal{M}_{\text{SFT}}$ consistently produces correct answers (i.e., $\hat{y}_i^n = y_i$ for all $n \in \{1, \ldots, N\}$), preference pairs are constructed based on the quality of the explanations. Let $s_i^n$ denote the score assigned by the judge $\mathcal{M}_{\text{Judge}}$ to the $n$-th explanation $\varepsilon_i^n$ for prompt $x_i$. We define two sets: $\mathbb{A}_i^w = \{\varepsilon_i^n : s_i^n = \max_{j \in \{1, \ldots, N\}} s_i^j\}$, which contains all explanations that achieve the highest score for prompt $x_i$, and $\mathbb{A}_i^l = \{\varepsilon_i^n : s_i^n < \max_{j \in \{1, \ldots, N\}} s_i^j\}$, which includes all explanations with scores lower than the maximum for prompt $x_i$.

**Preference Pairs for Variable Performance**: For input prompts $x_i$ where $\mathcal{M}_{\text{SFT}}$ produces a mix of correct and incorrect predictions (i.e., $\hat{y}_i^n \neq y_i$ for some $n \in \{1, \ldots, N\}$), preference pairs are constructed contrastively. We define the set $\mathbb{B}_i^w = \{\varepsilon_i^n : \hat{y}_i^n = y_i\}$, which contains explanations associated with correct predictions. From this set, we extract $\mathbb{A}_i^w \subseteq \mathbb{B}_i^w$, the subset of explanations with the highest scores assigned by $\mathcal{M}_{\text{Judge}}$, i.e., $\mathbb{A}_i^w = \{\varepsilon_i^n \in \mathbb{B}_i^w : s_i^n = \max_{j \in \mathbb{B}_i^w} s_i^j\}$. The set $\mathbb{A}_i^l = \{\varepsilon_i^n : \hat{y}_i^n \neq y_i \text{ and } s_i^n < \max_{j \in \mathbb{A}_i^w} s_i^j\}$ contains explanations corresponding to incorrect predictions, with scores lower than the maximum score in $\mathbb{A}_i^w$.

**Preference Pairs for Consistently Incorrect Prompts**: For prompts where all predictions from $\mathcal{M}_{\text{SFT}}$ are incorrect (i.e., $\hat{y}_i^n \neq y_i$ for all $n \in \{1, \ldots, N\}$), all corresponding explanations are placed in the set $\mathbb{A}_i^l$. To generate a winning explanation, we employ the $\mathcal{M}_{\text{Base}}$ model in a consultant role, similar to the LLM-as-a-Debater approach proposed by Khan et al. (2024). Since the inference hyperparameters for the LLM in this consulting role might differ from those used during the generation of preference pairs, we refer to this model as $\mathcal{M}_{\text{Debater}}$ to avoid confusion. Specifically, we provide the correct answer $y_i$ to the LLM and request an argument supporting this answer, which is then assigned to the set $\mathbb{A}_i^w$ as the winning explanation.

Finally, preference pairs are constructed for each instruction prompt $x_i$ by randomly sampling $\varepsilon_i^w$ from $\mathbb{A}_i^w$ as the winning explanation and $\varepsilon_i^l$ from $\mathbb{A}_i^l$ as the losing explanation. The resulting preference pair is denoted as $(x_i, \varepsilon_i^w, \varepsilon_i^l)$. The detailed algorithm is presented in Algorithm 1.

## 4 Experiments

In all experiments, we utilized `Llama-3-8B-Instruct` as our base model. Our study involved comparing four distinct model configurations:

1. $\mathcal{M}_{\text{Base}}$: The base model, which remains unmodified.

2. $\mathcal{M}_{\text{SFT}}$: This model was obtained by performing supervised fine-tuning on the $\mathcal{M}_{\text{Base}}$ model using only classification tasks, simulating scenarios where explanation annotations are not available.

---

**Algorithm 1** Generating Preference Pairs Via Anchor Selection

---

1: **Input:** Instruction prompt $x_i$, model predictions $\{\hat{y}_i^n\}_{n=1}^N$, true label $y_i$, judge model $\mathcal{M}_{\text{Judge}}$, debater model $\mathcal{M}_{\text{Debater}}$
2: **Output:** Preference pairs $(x_i, \varepsilon_i^w, \varepsilon_i^l)$
3: **Initialize:** $\mathbb{A}_i^w \leftarrow \emptyset$, $\mathbb{A}_i^l \leftarrow \emptyset$
4: **for** each explanation $\varepsilon_i^n$ **do**
5:      Compute score $s_i^n$ from $\mathcal{M}_{\text{Judge}}$
6: **end for**
7: **if** $\hat{y}_i^n = y_i$ for all $n \in \{1, \ldots, N\}$ **then**      $\triangleright$ Consistently Correct Prompts
8:      $\mathbb{A}_i^w \leftarrow \{\varepsilon_i^n : s_i^n = \max_{j \in \{1,\ldots,N\}} s_i^j\}$
9:      $\mathbb{A}_i^l \leftarrow \{\varepsilon_i^n : s_i^n = \min_{j \in \{1,\ldots,N\}} s_i^j\}$
10: **else if** $\hat{y}_i^n \neq y_i$ for some $n \in \{1, \ldots, N\}$ **then**      $\triangleright$ Variable Performance Prompts
11:      $\mathbb{B}_i^w \leftarrow \{\varepsilon_i^n : \hat{y}_i^n = y_i\}$
12:      $\mathbb{A}_i^w \leftarrow \{\varepsilon_i^n \in \mathbb{B}_i^w : s_i^n = \max_{j \in \mathbb{B}_i^w} s_i^j\}$
13:      $\mathbb{A}_i^l \leftarrow \{\varepsilon_i^n : \hat{y}_i^n \neq y_i \wedge s_i^n < \max_{j \in \mathbb{A}_i^w} s_i^j\}$
14: **else**      $\triangleright$ Consistently Incorrect Prompts
15:      $\mathbb{A}_i^l \leftarrow \{\varepsilon_i^n : \hat{y}_i^n \neq y_i$ for all $n \in \{1, \ldots, N\}\}$
16:      Generate argument $\varepsilon_i^{\text{debater}}$ using $\mathcal{M}_{\text{Debater}}$ given $y_i$
17:      $\mathbb{A}_i^w \leftarrow \{\varepsilon_i^{\text{debater}}\}$
18: **end if**
19: **Sample** $\varepsilon_i^w$ from $\mathbb{A}_i^w$
20: **Sample** $\varepsilon_i^l$ from $\mathbb{A}_i^l$
21: **Return** $(x_i, \varepsilon_i^w, \varepsilon_i^l)$

---

3. $\mathcal{M}_{\text{Rank}}$: The $\mathcal{M}_{\text{SFT}}$ model was further refined using DPO, employing a self-preference dataset constructed from rank-ordered preference pairs derived solely from judge-based evaluations of the explanations. This approach aligns with methodologies described in Bai et al. (2022); Wang et al. (2023); Yuan et al. (2024); Wu et al. (2024).

4. $\mathcal{M}_{\text{Anchor}}$ (ours): Similar to $\mathcal{M}_{\text{Rank}}$, this model was refined using DPO but utilized a self-preference dataset created with our proposed anchored method for selecting preference pairs.

Since both $\mathcal{M}_{\text{Rank}}$ and $\mathcal{M}_{\text{Anchor}}$ undergo an additional stage of DPO alignment with the self-preference dataset, we will collectively refer to them as self-aligned models in comparison to $\mathcal{M}_{\text{SFT}}$.

### 4.1 EXPERIMENTAL SETUP

**Datasets**: We selected four datasets for our experiments: `AQuA-Rat` (Ling et al., 2017), `ARC-Challenge` (Clark et al., 2018), `LogiQA` (Liu et al., 2020), and `OpenbookQA` (Mihaylov et al., 2018). These datasets are established benchmarks for reasoning tasks, requiring a challenging reasoning process, which makes them an ideal fit for evaluating the quality of self-explanations. A key factor in their selection was the size of their training sets, which provided a sufficient number of input prompts to support the creation of the self-preferenceion dataset. In the case of `AQuA-Rat`, we sampled 5,000 examples due to computational constraints. For evaluation, we used the test split of each dataset. For detailed dataset descriptions, see Appendix D.

**SFT Training Details**: For $\mathcal{M}_{\text{SFT}}$, we used the AdamW optimizer with a learning rate of $5 \times 10^{-5}$ for one epoch, following a cosine schedule with 10% warmup steps. Gradient clipping was set to 0.3, and we used an effective batch size of 12. Loss was computed only on the assistant's completions. Instead of fine-tuning the entire model, we applied a LoRA adapter ($\alpha = 128$, dropout = 0.05, rank $r = 256$) to all linear layers. LoRA adapters were used to accelerate training and to act as a regularization method (Biderman et al., 2024), addressing the overfitting tendencies of DPO (Thakkar et al., 2024), which is applied during the later alignment phase.

**Self-Preference Dataset**: To ensure the integrity of our evaluation process, we created separate self-preference datasets for each benchmark. These datasets were built using input prompts specific to

each task, ensuring that the DPO alignment data remained uncontaminated by exposure to multiple tasks. This approach prevents the artificial inflation of results that could occur if models were aligned across diverse tasks, unlike SFT models, which are trained on one classification task at a time. To create the self-preference dataset for aligning $\mathcal{M}_{\text{Rank}}$ and $\mathcal{M}_{\text{Anchor}}$, we sampled $N = 4$ responses from $\mathcal{M}_{\text{SFT}}$ for each input prompt (settings: temperature $T = 0.6$ and top-k value of 0.9). This sample size provided a reasonable assessment of the model's consistency and variability. The prompt used is presented in Appendix G. In cases of consistently incorrect responses, $\mathcal{M}_{\text{Base}}$ was employed as the debater model (`Llama-3-8B-Instruct`) with adjusted parameters ($T = 0.5$, top-k 0.9). The responses were scored by $\mathcal{M}_{\text{Judge}}$, which was the same base model, ensuring a self-contained alignment process. This setup differs from the evaluation phase, where a more capable model serves as the judge. The scoring of explanations followed the methodology outlined in Section 2.2, with $\mathcal{M}_{\text{Judge}}$ using fixed inference parameters ($T = 0$). For $\mathcal{M}_{\text{Rank}}$, preference pairs were chosen based on the assigned scores, with the highest-scoring explanation designated as the winner, and the losing explanation randomly selected from the remaining candidates. The preference pairs for $\mathcal{M}_{\text{Anchor}}$ were selected using the methodology outlined in Section 3.3, and these pairs were then used to align the models through DPO.

**DPO Training Details**: For DPO-aligned models ($\mathcal{M}_{\text{Rank}}$, $\mathcal{M}_{\text{Anchor}}$), we used similar hyperparameters as in the SFT phase but reduced the learning rate to $5 \times 10^{-7}$ and trained for 2.6k steps with an effective batch size of 6. The DPO process used a $\beta$ value of 0.1 and updated the LoRA weights obtained during SFT.

**Evaluation**: We evaluated our models along two main dimensions: prediction accuracy and self-explanation quality. To account for variability in model outputs, we generated $N = 16$ explanation-prediction pairs per input prompt. The inference settings mirrored those used to create the self-preference dataset, with a temperature of $T = 0.6$ and top-k set to 0.9 and the same prompt used during the creation of the self-preference dataset (see Appendix G.3). Average prediction accuracy was used to measure performance on downstream tasks. To assess self-explanation quality, we performed head-to-head comparisons between the aligned models ($\mathcal{M}_{\text{SFT}}$, $\mathcal{M}_{\text{Rank}}$, $\mathcal{M}_{\text{Anchor}}$) and the base model ($\mathcal{M}_{\text{Base}}$).

**Ablation Study**: To validate our design choices, we conduct two types of ablation experiments. First, we create variants of $\mathcal{M}_{\text{Anchor}}$ by combining different strategies outlined in Section 3.3 to construct the anchor-preference dataset. The variants studied include $\mathcal{M}_{\text{Anchor (CC)}}$, $\mathcal{M}_{\text{Anchor (CC+V)}}$, and $\mathcal{M}_{\text{Anchor (CC+V+CI)}}$, where "Consistently Correct" (CC), "Variant" (V), and "Consistently Incorrect" (CI) denote the respective strategies. Notably, $\mathcal{M}_{\text{Anchor (CC+V+CI)}}$ represents the full version of $\mathcal{M}_{\text{Anchor}}$, and the terms may be used interchangeably throughout the discussion.

Furthermore, we evaluate the aligned models using `Mistral-Large-Instruct-2407`, `Qwen2.5-72B-Instruct`, and `Llama3-70B-Instruct` as $\mathcal{M}_{\text{Judge}}$, validating the robustness of our method across diverse evaluation settings.

### 4.1.1 IMPACT OF SUPERVISED FINE-TUNING ON SELF-EXPLANATIONS

We observed a significant trade-off in evaluation results before and after applying supervised fine-tuning on a classification task (see Table 1). While SFT notably improved classification accuracy, it resulted in a substantial decline in the quality of self-explanations compared to the base model. The average decrement in $\mathcal{M}_{\text{SFT}}$ from the equilibrium point, as measured across judges, ranged from 6.5% to 13.3% across benchmarks.

Building on evidence that supervised fine-tuning can improve performance on specific tasks at the expense of a model's generalization abilities (Yang et al., 2024; Kirk et al., 2024), we hypothesize that this decline occurs because the task of selecting predefined answers does not inherently encourage the model to articulate its reasoning, leading to a specialization that diminishes the quality of the generated explanations.

These findings highlight the necessity for alignment techniques that can preserve high-quality explanations in situations where datasets with annotated explanations are unavailable for fine-tuning.

Table 1: **Comparison of Aligned Models.** The table presents average accuracy alongside pairwise evaluations of self-explanation quality. Pairwise evaluations are conducted using `Mistral-Large-Instruct-2407`, `Qwen2.5-72B-Instruct`, and `LLama3-70B-Instruct` as judges. Additionally, for the ablation study, we report the performance of $\mathcal{M}_{\text{Anchor}}$ under various strategy combinations used to construct the self-preference dataset. The strategies include Consistently Correct (CC), Variant (V), and Consistently Incorrect (CI).

| Dataset | $\mathcal{M}_{\textbf{Base}}$ **Acc. (%)** | $\mathcal{M}_{\textbf{Align}}$ **Type** | $\mathcal{M}_{\textbf{Align}}$ **Acc. (%)** | $\varepsilon \left(W + \frac{T}{2}\right)$ **Rate (%)** $\uparrow$ | | | |
|---|---|---|---|---|---|---|---|
| | | | | $J_{\text{LLama}}$ | $J_{\text{Qwen}}$ | $J_{\text{Mistral}}$ | $J_{\text{Avg}}$ |
| AQuA Rat | $47.1_{\pm 2.9}$ | $\mathcal{M}_{\text{SFT}}$ | $47.7_{\pm 2.7}$ | 42.2 | 47.2 | 41.1 | 43.5 |
| | | $\mathcal{M}_{\text{Rank (Baseline)}}$ | $48.3_{\pm 2.1}$ | 43.1 | 48.3 | 41.5 | 44.3 |
| | | $\mathcal{M}_{\text{Anchor (CC)}}$ | $49.3_{\pm 3.1}{}^{*}$ | 44.7 | 49.0 | 43.3 | 45.7 |
| | | $\mathcal{M}_{\text{Anchor (CC+V)}}$ | $48.3_{\pm 2.9}$ | 42.9 | 48.1 | 42.6 | 44.5 |
| | | $\mathcal{M}_{\text{Anchor (CC+V+CI)}}$ | $\mathbf{51.1}_{\pm 3.0}$ | **48.0** | **49.5** | **46.3** | **47.9** |
| ARC Challenge | $76.4_{\pm 0.7}$ | $\mathcal{M}_{\text{SFT}}$ | $81.0_{\pm 0.7}$ | 36.8 | 32.0 | 41.3 | 36.7 |
| | | $\mathcal{M}_{\text{Rank (Baseline)}}$ | $81.9_{\pm 1.1}{}^{*}$ | 48.4 | 48.2 | 49.2 | 48.6 |
| | | $\mathcal{M}_{\text{Anchor (CC)}}$ | $81.6_{\pm 1.3}{}^{*}$ | 47.3 | 46.7 | 48.0 | 47.3 |
| | | $\mathcal{M}_{\text{Anchor (CC+V)}}$ | $\mathbf{82.0}_{\pm 1.1}$ | 48.6 | 48.7 | 49.7 | 49.0 |
| | | $\mathcal{M}_{\text{Anchor (CC+V+CI)}}$ | $\mathbf{82.0}_{\pm 0.9}$ | **51.6** | **52.1** | **52.4** | **52.0** |
| LogiQA | $41.4_{\pm 1.1}$ | $\mathcal{M}_{\text{SFT}}$ | $45.2_{\pm 0.7}$ | 34.6 | 34.6 | 42.6 | 37.3 |
| | | $\mathcal{M}_{\text{Rank (Baseline)}}$ | $46.0_{\pm 1.5}{}^{*}$ | 47.1 | 45.0 | 47.8 | 46.6 |
| | | $\mathcal{M}_{\text{Anchor (CC)}}$ | $45.8_{\pm 1.4}{}^{*}$ | 48.5 | 46.8 | 49.2 | 48.2 |
| | | $\mathcal{M}_{\text{Anchor (CC+V)}}$ | $46.1_{\pm 1.7}{}^{*}$ | 47.3 | 45.2 | 48.1 | 46.9 |
| | | $\mathcal{M}_{\text{Anchor (CC+V+CI)}}$ | $\mathbf{46.6}_{\pm 2.2}$ | **53.4** | **50.9** | **51.3** | **51.9** |
| Openbook QA | $71.7_{\pm 1.3}$ | $\mathcal{M}_{\text{SFT}}$ | $\mathbf{87.4}_{\pm 1.1}$ | 38.4 | 36.2 | 46.3 | 40.3 |
| | | $\mathcal{M}_{\text{Rank (Baseline)}}$ | $87.0_{\pm 1.1}{}^{*}$ | 45.4 | 45.1 | 48.9 | 46.5 |
| | | $\mathcal{M}_{\text{Anchor (CC)}}$ | $87.1_{\pm 0.5}{}^{*}$ | 45.5 | 45.4 | 49.2 | 46.7 |
| | | $\mathcal{M}_{\text{Anchor (CC+V)}}$ | $\mathbf{87.4}_{\pm 0.8}$ | 46.2 | 46.0 | **49.6** | 47.3 |
| | | $\mathcal{M}_{\text{Anchor (CC+V+CI)}}$ | $87.0_{\pm 0.9}{}^{*}$ | **46.5** | **46.9** | **49.6** | **47.7** |

### 4.1.2 ANALYSIS OF SELF-ALIGNED MODELS

**Prediction Accuracy:** The self-preference dataset used during the DPO phase is designed with the primary objective of improving explanation quality rather than maximizing accuracy. However, it is essential to ensure that enhancing the models' self-explanation capabilities does not compromise their performance on the primary task, as measured by classification accuracy. To assess this, we compute the average classification accuracy[1] and the standard deviation across multiple evaluation runs ($N = 16$), as shown in Table 1.

The results indicate that the self-aligned models, $\mathcal{M}_{\text{Rank}}$ and $\mathcal{M}_{\text{Anchor}}$, across all tested strategy combinations, either maintain or improve upon the classification accuracy gains achieved by the seed model, $\mathcal{M}_{\text{SFT}}$, relative to the base model, $\mathcal{M}_{\text{Base}}$. Notably, while the accuracy performances of $\mathcal{M}_{\text{Anchor}}$ and $\mathcal{M}_{\text{Rank}}$ are similar across most tasks, there is one exception: $\mathcal{M}_{\text{Anchor}}$ significantly outperforms $\mathcal{M}_{\text{Rank}}$ on the `Aqua-Rat` dataset, with statistical significance. Further analysis of the variability in $\mathcal{M}_{\text{Anchor}}$'s performance across different datasets is provided in Section 4.1.4.

**Self-Explanation Quality**: Pairwise evaluations of self-explanation quality (see Table 1) show that the initial decline in explanation performance observed in $\mathcal{M}_{\text{SFT}}$ is partially inherited by both $\mathcal{M}_{\text{Rank}}$ and $\mathcal{M}_{\text{Anchor}}$, since they both use $\mathcal{M}_{\text{SFT}}$ as the seed model during the DPO alignment phase. Nevertheless, both $\mathcal{M}_{\text{Rank}}$ and $\mathcal{M}_{\text{Anchor}}$ achieve significant improvements in explanation quality over $\mathcal{M}_{\text{SFT}}$, with $\mathcal{M}_{\text{Anchor}}$ demonstrating the strongest performance across benchmarks and evaluation judges. When compared to the base model, $\mathcal{M}_{\text{Anchor}}$ shows similar performance, winning half of the

---

[1]The maximum value is bolded, and results marked with a (*) indicate no statistical difference from the top performer.

benchmarks, and significantly narrows the gap in explanation quality introduced by $\mathcal{M}_{\text{SFT}}$ on the remaining benchmark datasets. Regarding the ablation study, we observe that the highest explanation quality is achieved when the strategies (CC), (V), and (CI) are combined.

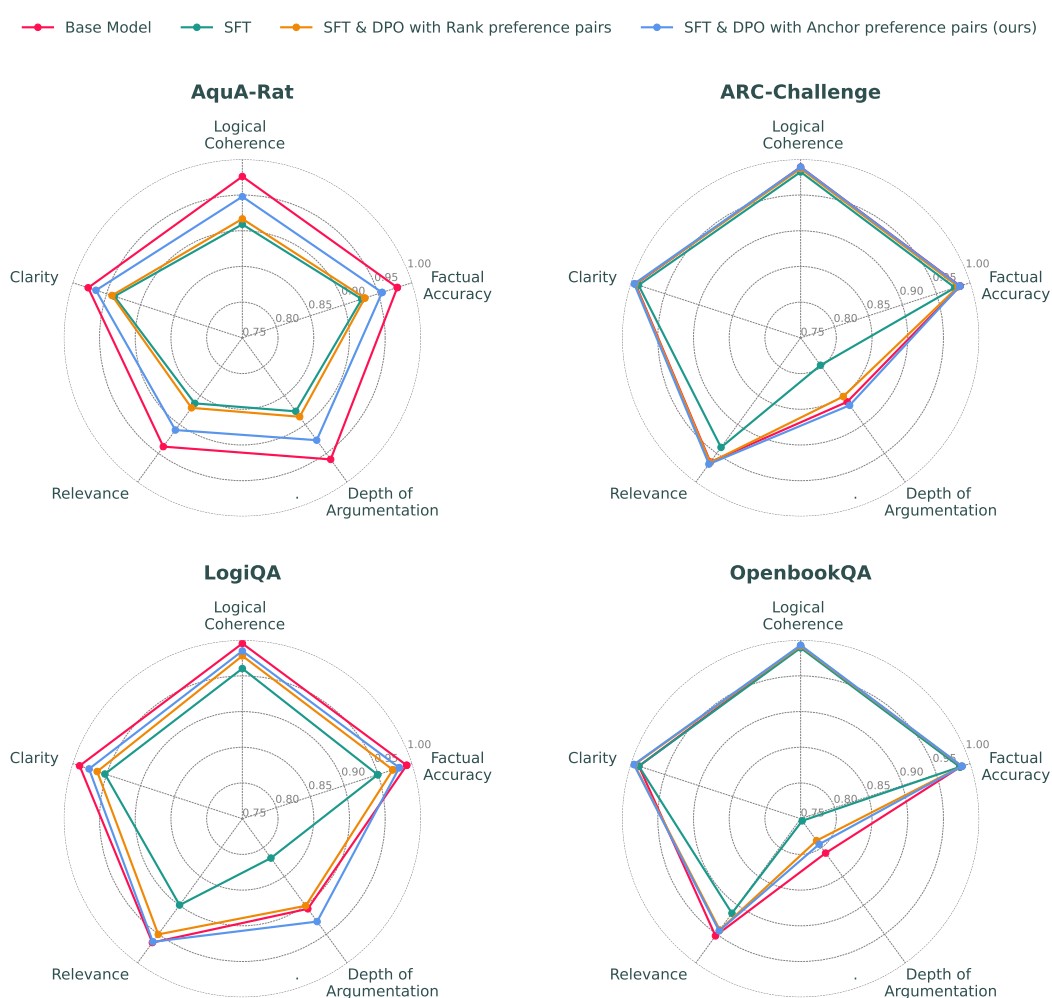

Figure 1: **Average Self-Explanation Scores per Evaluation Criterion**. Average scores for each evaluation criterion used to assess self-explanations, as described in Section 2.1. The scores are provided for all evaluated models across the benchmark datasets.

### 4.1.3 ANALYSIS OF INDIVIDUAL EVALUATION DIMENSIONS

Figure 1 presents the average scores for each evaluation criterion used to assess self-explanations, as described in Section 2.1, for all evaluated models across the benchmark datasets.

Overall, the self-aligned models outperform $\mathcal{M}_{\text{SFT}}$ across all evaluation criteria, with $\mathcal{M}_{\text{Anchor}}$ consistently achieving better results than $\mathcal{M}_{\text{Rank}}$.

Additionally, we observe that the degradation in self-explanation quality due to SFT varies significantly depending on the dataset used for fine-tuning. Two notable trends emerge from the analysis. First, for more complex tasks—where complexity is measured by lower test accuracy—such as AQuA-Rat and LogiQA, the decline in explanation quality is more pronounced across all criteria. Second, evaluation dimensions for which the base model originally received lower scores tend to experience a more significant drop in performance after SFT.

### 4.1.4 Impact of Preference Pairs Category Distribution

We define $\lambda$ as the proportion of the self-preference dataset used to align $\mathcal{M}_{\text{Anchor}}$, corresponding to preference pairs selected under the (CI) or (V) strategies (see Section 3.3). This metric, $\lambda$, offers insight into how the composition of the self-preference dataset for $\mathcal{M}_{\text{Anchor}}$ differs from that of $\mathcal{M}_{\text{Rank}}$.

The (CI) and (V) strategies ensure—assuming the self-explanation is faithful—that the winning explanation $\varepsilon_i^w$ supports the ground-truth label $y_i$. For the (V) strategy, this is achieved by sampling $\varepsilon_i^w$ from the set $\mathbb{B}_i^w = \{\varepsilon_i^n : \hat{y}_i^n = y_i\}$. In the case of the (CI) strategy, $\mathcal{M}_{\text{Debater}}$ is employed to provide arguments that explicitly support $y_i$.

In contrast, $\mathcal{M}_{\text{Rank}}$—which serves as our baseline—selects pairs for the preference dataset solely based on scores assigned by judges during its creation. As a result, it does not guarantee that the winning explanations, $\varepsilon_i^w$, will support the ground-truth label $y_i$. The likelihood of selecting cases in the preference dataset where $\varepsilon_i^w$ aligns with an outcome different from $y_i$ increases, particularly as $\lambda$ grows.

We evaluated improvements by analyzing the differences in accuracy and $J_{\text{Avg}}$ between $\mathcal{M}_{\text{Anchor}}$ and $\mathcal{M}_{\text{Rank}}$ in relation to $\lambda$ (see Figure 2). In both cases, we observed a trend showing that $\mathcal{M}_{\text{Anchor}}$ demonstrates a greater relative improvement compared to $\mathcal{M}_{\text{Rank}}$ as $\lambda$ increases. This supports our design principle that tailoring strategies based on model behavior is crucial for improving the quality of self-preference datasets and avoiding the reinforcement of problematic behavior.

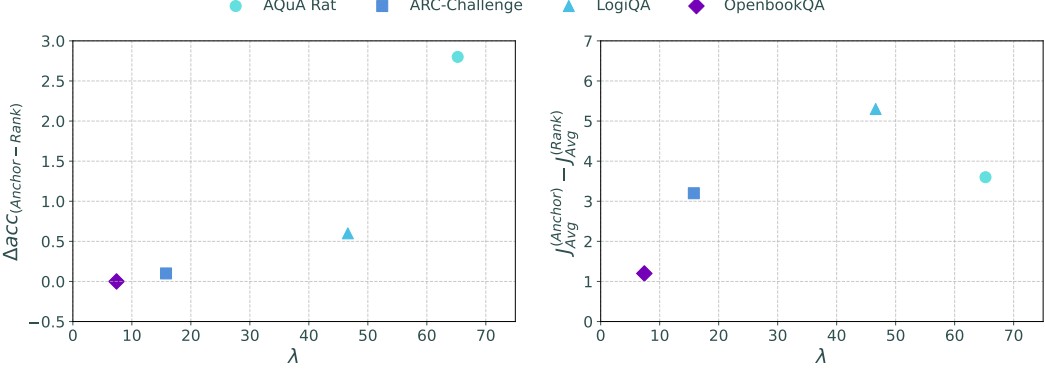

Figure 2: **Impact of Preference Pairs Category Distribution:** Presents the relative improvements in accuracy (*left*) and $J_{\text{Avg}}$ difference between $\mathcal{M}_{\text{Anchor}}$ and $\mathcal{M}_{\text{Rank}}$ (*right*) with respect to $\lambda$.

## 5 Related Work

**LLM-as-Evaluator**: This concept refers to the ability of large language models (LLMs) to evaluate the outputs of other LLMs, a technique commonly referred to as LLM-as-a-Judge. This approach has gained considerable traction in recent years (Dubois et al., 2023; Li et al., 2024; Fernandes et al., 2023; Bai et al., 2023) and is frequently used to assess LLM performance across various downstream tasks (Zheng et al., 2023). It has proven particularly effective in automating evaluations, as demonstrated on platforms like LMSys Chatbot Arena. Key implementations include direct scoring based on specific criteria, pairwise comparisons (Liu et al., 2024), reference-based evaluations, and ensemble methods (Verga et al., 2024). While LLM-as-a-Judge offers scalability and consistency, it can also inherit biases from the evaluation model, potentially amplifying problematic outputs (Huang et al., 2024). Despite these challenges, it remains a valuable tool due to its efficiency and cost-effectiveness in evaluating LLM systems. In our work, we introduce a framework for the qualitative assessment of *self-explanations* using the LLM-as-a-Judge technique, designed to evaluate how effectively a model conveys its reasoning.

**Self-Alignment**: Several approaches have been developed to improve LLMs without requiring human-annotated feedback. One method involves fine-tuning models using high-quality, self-

generated input-output pairs (Wang et al., 2023; Chen et al., 2023; Gulcehre et al., 2023), though this can perpetuate biases in example selection without a clear mechanism for improving selection quality. Another influential approach is Constitutional AI (Bai et al., 2022), where an LLM provides feedback and refines responses, which are then used to train a separate, static reward model. Building on this concept, Yuan et al. (2024) and Wu et al. (2024) proposed using the LLM itself as a dynamic reward model, eliminating the need for a static one. This allows for continuous improvement in both generation and evaluation capabilities through iterative training processes. In our work, we introduce a novel method for creating a self-preference dataset. Our approach, called *Alignment with Anchor Preference Pairs*, enhances preference pair selection by categorizing model behavior in response to each input prompt and applying tailored strategies for each category. To evaluate a model's consistency for a given input prompt, an anchor—i.e., a ground-truth reference—is required, which we derive from a classification task used as a probing mechanism.

**LLM-as-a-Debater**: This adversarial approach aims to improve model performance through argumentation. In Perez et al. (2019), debaters are limited to extracting relevant statements from a source text, rather than generating original arguments. Du et al. (2023) extended this concept by involving multiple LLM instances to debate their individual responses over several rounds, eventually converging on a shared final answer. Khan et al. (2024) further developed this approach by using debate-like scenarios to challenge and refine model outputs through simulated arguments. In our work, we adopt the LLM-as-a-Debater approach in the role of a consultant, specifically following Khan et al. (2024), for cases where the model's response to certain input prompts is *consistently incorrect*. This strategy enables the creation of self-preference examples that avoid reinforcing problematic behavior.

## 6 LIMITATIONS

We acknowledge some limitations in our approach. First, evaluating the model's consistency on a given input prompt requires a anchor—ground truth reference. Consequently, the selection of preference pairs via the anchor strategy relies on a classification task as the probing mechanism, which restricts its applicability. Second, when ranking the quality of self-explanations, we assign equal weights across all evaluation dimensions. This uniform weighting may not accurately reflect the varying significance of different aspects of explanation quality, which can differ depending on the user or specific application. Moreover, this approach may overlook instances where individual explanations degrade in separate criteria, potentially leading to preference pairs where score differences arise from unrelated factors.

Finally, using the base model as the judge during the creation of the self-preference dataset addresses the issue that $\mathcal{M}_{\text{STF}}$ exhibits a decline in explanation quality (see Section 4.1.1) and reduced agreement with the evaluation judge (see Appendix C). However, this approach could become a bottleneck in scenarios where multiple iterations of alignment are performed, as the base model remains static and may fail to capture important evaluation nuances over time.

## 7 CONCLUSION

In this work, we introduced a methodology for alignment that enhances LLMs' ability to generate high-quality self-explanations, even in the absence of annotated rationale explanations. Our approach provides an end-to-end solution for aligning LLMs on classification tasks, ensuring that they not only produce accurate predictions but also articulate coherent explanations for their decisions. This is achieved through three core components: evaluating the quality of generated explanations, creating self-preference datasets, and aligning the model. Central to our approach is Alignment with Anchor Preference Pairs, a novel method that refines preference pair selection by categorizing model outputs into three groups: consistently correct, consistently incorrect, and variable. For each category, we apply tailored strategies to construct preference pairs, which are then used in DPO. Our empirical results demonstrate that this method consistently improves explanation quality, reducing the degradation caused by task specialization. Furthermore, we show that using anchor preference pairs outperforms self-alignment methods that rely solely on judge-based evaluations for preference pair selection.

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

## A  PAIRWISE MODEL EVALUATION

To compare the performance of two models, denoted as $\mathcal{M}_1$ and $\mathcal{M}_2$, we perform a pairwise evaluation of the self-explanations generated for a given prompt $x_i$. Each model produces $N$ explanations, and we compare each explanation from $\mathcal{M}_1$ with every explanation from $\mathcal{M}_2$, resulting in $N^2$ pairwise comparisons.

For a given comparison between the $n$-th explanation from model $\mathcal{M}_1$ and the $m$-th explanation from model $\mathcal{M}_2$, where $n, m \in \{1, \ldots, N\}$, we compare the corresponding scores, $s_i^n(\mathcal{M}_1)$ and $s_i^m(\mathcal{M}_2)$. A *win* for $\mathcal{M}_1$ is recorded if the score from $\mathcal{M}_1$ is strictly greater than that from $\mathcal{M}_2$:

$$s_i^n(\mathcal{M}_1) > s_i^m(\mathcal{M}_2)$$

Conversely, a *loss* for $\mathcal{M}_1$ occurs if the score from $\mathcal{M}_1$ is strictly less than the score from $\mathcal{M}_2$:

$$s_i^n(\mathcal{M}_1) < s_i^m(\mathcal{M}_2)$$

A *tie* is defined when both scores are equal:

$$s_i^n(\mathcal{M}_1) = s_i^m(\mathcal{M}_2)$$

For each prompt $x_i$, we count the total number of wins, losses, and ties across all $N^2$ comparisons between the explanations from both models. To summarize the performance of the models across the entire dataset, we compute the win rate, tie rate, and loss rate.

The win rate $W(\mathcal{M}_1, \mathcal{M}_2)$ is the average proportion of pairwise comparisons in which model $\mathcal{M}_1$ outperforms model $\mathcal{M}_2$ across all prompts in the set $\mathcal{X}$. It is computed as:

$$W(\mathcal{M}_1, \mathcal{M}_2) = \frac{1}{|\mathcal{X}|} \sum_{x_i \in \mathcal{X}} \left( \frac{1}{N^2} \sum_{n=1}^{N} \sum_{m=1}^{N} \mathbb{1}[s_i^n(\mathcal{M}_1) > s_i^m(\mathcal{M}_2)] \right)$$

Here, $\mathcal{X}$ is the set of all prompts, and $\mathbb{1}[\cdot]$ is the indicator function, which returns 1 if the condition inside the brackets is true (i.e., if $\mathcal{M}_1$ wins) and 0 otherwise.

Similarly, we define the tie rate $T(\mathcal{M}_1, \mathcal{M}_2)$ as the proportion of pairwise comparisons where the models perform equally:

$$T(\mathcal{M}_1, \mathcal{M}_2) = \frac{1}{|\mathcal{X}|} \sum_{x_i \in \mathcal{X}} \left( \frac{1}{N^2} \sum_{n=1}^{N} \sum_{m=1}^{N} \mathbb{1}[s_i^n(\mathcal{M}_1) = s_i^m(\mathcal{M}_2)] \right)$$

The loss rate $L(\mathcal{M}_1, \mathcal{M}_2)$ captures the proportion of comparisons where $\mathcal{M}_1$ performs worse than $\mathcal{M}_2$:

$$L(\mathcal{M}_1, \mathcal{M}_2) = \frac{1}{|\mathcal{X}|} \sum_{x_i \in \mathcal{X}} \left( \frac{1}{N^2} \sum_{n=1}^{N} \sum_{m=1}^{N} \mathbb{1}[s_i^n(\mathcal{M}_1) < s_i^m(\mathcal{M}_2)] \right)$$

Table 2: **Inter-Rater Reliability Scores for Human Evaluators:** This table presents Krippendorff's $\alpha$ coefficients across five evaluation criteria, measuring the degree of agreement among human raters.

| Criteria | Krippendorff's $\alpha$ |
|---|---|
| Factual Accuracy | 0.25 |
| Logical Coherence | 0.16 |
| Clarity and Comprehensibility | 0.24 |
| Relevance and Completeness | 0.20 |
| Depth of Argumentation | 0.47 |

By evaluating the win, tie, and loss rates, we obtain a comprehensive picture of how the two models compare in terms of generating higher-quality self-explanations. This approach accounts for variability in the quality of explanations generated by the models, providing a more nuanced and detailed evaluation than methods relying solely on single-point estimates. Throughout the evaluations presented in this work, $\mathcal{M}_2$ refers to the baseline model $\mathcal{M}_{\text{Base}}$.

## B CORRELATION WITH HUMAN JUDGMENTS

We conducted a human evaluation by sampling 30 explanations generated by either an aligned model or the base model across 10 distinct questions sourced from the `LogiQA` dataset, which features a wide range of logical reasoning problems across different domains. Three state-of-the-art language models: `Mistral-Large-Instruct-2407`, `Qwen2.5-72B-Instruct`, and `LLama3-70B-Instruct` served as automated judges, rating explanations on a five-point scale: `Bad`, `Poor`, `Fair`, `Good`, `Excellent`.

In addition, the same set of questions were included in a survey administered to multiple human raters. Each rater independently evaluated the quality of the explanations based on a specific criterion (e.g., Depth of Argumentation). To ensure diversity and reliability in the evaluations, each explanation was rated by at least three human raters.

### B.1 METRIC: KRIPPENDORFF'S $\alpha$

We computed Inter-Rater Reliability (IRR) among human raters using Krippendorff's $\alpha$ for each evaluation criterion. The results, presented in Table 2, indicate fair agreement overall, with some notable variations: moderate agreement for Depth of Argumentation and slight agreement for Logical Coherence.

Our results in Table 2 reveal varying levels of inter-rater agreement. **Depth of Argumentation** achieved the strongest consensus with a Krippendorff's $\alpha$ of 0.47 (moderate), indicating consistent assessment of explanation thoroughness. **Factual Accuracy** and **Clarity and Comprehensibility** showed fair agreement ($\alpha = 0.25$ and $0.24$ respectively), suggesting some alignment in evaluating correctness and clarity while highlighting notable rater variability. **Logical Coherence** had the weakest agreement ($\alpha = 0.16$), demonstrating substantial differences in how raters evaluated logical flow.

### B.2 METRIC: WEIGHTED COHEN'S KAPPA

We generated a consensus rating for each item by calculating the median of human ratings, considering the ordinal scale of the evaluation. Agreement between the LLMs' ratings and the human consensus ratings was measured using Weighted Cohen's Kappa, which penalizes disagreements proportionally to the ordinal distance. The results are presented in Table 3.

Table 3 demonstrates varying degrees of model-human agreement in evaluation criteria. `Mistral-Large-Instruct-2407` exhibited the strongest alignment, with Weighted Kappa values between 0.38-0.49. The model achieved notably high scores in Depth of Argumentation (0.49), Factual Accuracy (0.48), and Relevance and Completeness (0.48). These values, falling

within the moderate agreement range (0.41-0.60), demonstrate the model's robust ability to replicate human evaluation patterns.

`Qwen2.5-72B-Instruct` showed fair agreement with human consensus, with Kappa values spanning 0.28-0.42. The model performed best in assessing Factual Accuracy (0.42) and Depth of Argumentation (0.37), though its alignment remained less consistent than Mistral's.

`LLama3-70B-Instruct` demonstrated the weakest alignment, with most Kappa values below 0.35. Its peak performance in Relevance and Completeness (0.34) and Depth of Argumentation (0.31) barely reached fair agreement thresholds, indicating substantial divergence from human evaluation patterns.

### B.3 METRIC: SPEARMAN CORRELATION

Additionally, we calculated the Spearman correlation between the LLMs' ratings and the human consensus ratings to evaluate the strength and direction of the monotonic relationship between the two variables. The results are presented in Table 3.

The Spearman correlation analysis revealed moderate positive correlations across all three models, with most coefficients ranging from 0.28 to 0.52. `Mistral-Large-Instruct-2407` demonstrated the strongest correlations, with coefficients consistently above 0.45 across all criteria ($p < 0.01$). Particularly notable were its correlations in Clarity ($\rho = 0.52$), Factual Accuracy ($\rho = 0.51$), and Depth of Argumentation ($\rho = 0.50$), indicating reliable alignment with human judgments.

`Qwen2.5-72B-Instruct` showed more variable performance, with correlations ranging from weak to moderate. While it achieved significant correlations in Clarity ($\rho = 0.48$, $p < 0.01$) and Factual Accuracy ($\rho = 0.45$, $p < 0.02$), its correlation for Relevance was notably weaker and non-significant ($\rho = 0.28$, $p = 0.15$).

`LLama3-70B-Instruct` exhibited moderate correlations across all criteria ($\rho = 0.39$–$0.50$), with all relationships reaching statistical significance ($p < 0.05$). Its strongest correlation was in Depth of Argumentation ($\rho = 0.50$, $p < 0.01$), matching Mistral's performance in this category.

Figure 3 visualizes the relationship between LLM ratings and human consensus scores across five evaluation criteria. Each point represents the mean LLM rating for a given human consensus score, with error bars indicating standard deviation. The plots reveal systematic patterns in how different LLMs interpret and rate explanations compared to human evaluators. Notably, (a) Mistral-Large-2407 shows the most consistent alignment with human judgments, particularly in Factual Accuracy and Depth of Argumentation, (b) Qwen2.5-72B-Instruct demonstrates moderate alignment but with higher variance in middle-range scores, and (c) LLama3-70B-Instruct exhibits a tendency to overestimate ratings, especially for lower human consensus scores.

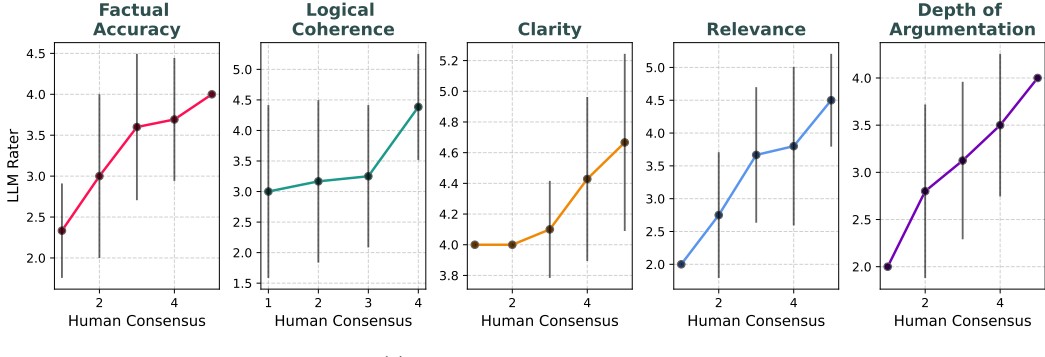

(a) `Mistral-Large-2407`

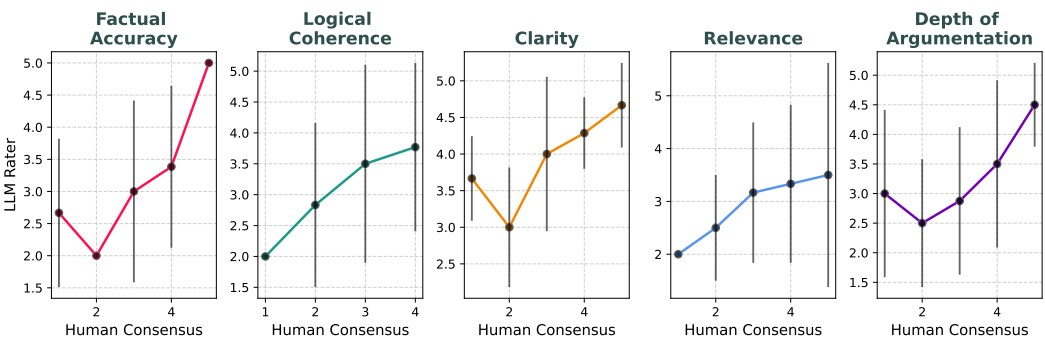

(b) `Qwen2.5-72B-Instruct`

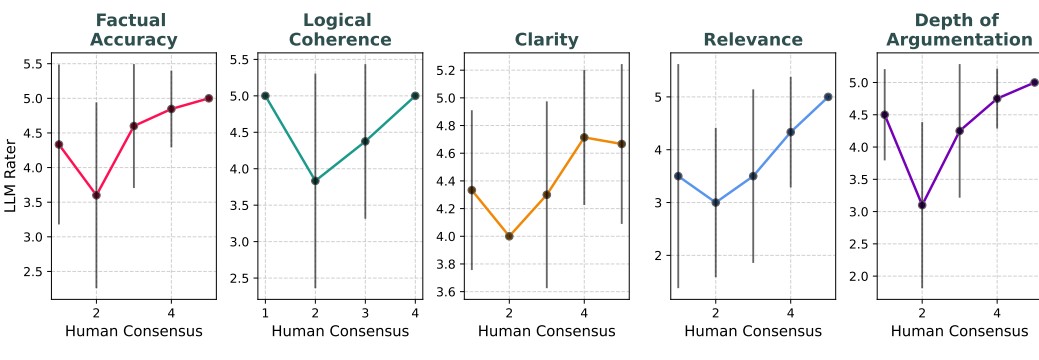

(c) `LLama3-70B-Instruct`

Figure 3: **LLM vs Human Consensus Ratings Across Evaluation Criteria:** Mean LLM ratings plotted against human consensus scores for each evaluation dimension with standard deviation error bars.

Table 3: **Analysis of Agreement Between LLM and Human Evaluations:** A comparative analysis of agreement between LLM-based and human consensus ratings using two metrics: Weighted Cohen's $\kappa$ (measuring absolute agreement) and Spearman Correlation (measuring ranking consistency). Results are broken down by evaluation criteria for each LLM judge, with $p$-values indicating statistical significance ($*p < 0.05$).

| LLM-as-a-Judge | Criteria | Metric | Value | $p$-value |
|---|---|---|---|---|
| Mistral-Large Instruct-2407 | Factual Accuracy | Weighted Cohen's $\kappa$ | 0.48 | - |
| | | Spearman Correlation | 0.51 | 0.01* |
| | Logical Coherence | Weighted Cohen's $\kappa$ | 0.38 | - |
| | | Spearman Correlation | 0.49 | 0.01* |
| | Clarity | Weighted Cohen's $\kappa$ | 0.17 | - |
| | | Spearman Correlation | 0.52 | 0.01* |
| | Relevance | Weighted Cohen's $\kappa$ | 0.48 | - |
| | | Spearman Correlation | 0.45 | 0.01* |
| | Depth of Argumentation | Weighted Cohen's $\kappa$ | 0.49 | - |
| | | Spearman Correlation | 0.50 | 0.01* |
| Qwen2.5 72B-Instruct | Factual Accuracy | Weighted Cohen's $\kappa$ | 0.42 | - |
| | | Spearman Correlation | 0.45 | 0.02* |
| | Logical Coherence | Weighted Cohen's $\kappa$ | 0.32 | - |
| | | Spearman Correlation | 0.34 | 0.07* |
| | Clarity | Weighted Cohen's $\kappa$ | 0.32 | - |
| | | Spearman Correlation | 0.48 | 0.01* |
| | Relevance | Weighted Cohen's $\kappa$ | 0.28 | - |
| | | Spearman Correlation | 0.28 | 0.15 |
| | Depth of Argumentation | Weighted Cohen's $\kappa$ | 0.37 | - |
| | | Spearman Correlation | 0.39 | 0.04* |
| LLama3-70B Instruct | Factual Accuracy | Weighted Cohen's $\kappa$ | 0.20 | - |
| | | Spearman Correlation | 0.41 | 0.03* |
| | Logical Coherence | Weighted Cohen's $\kappa$ | 0.14 | - |
| | | Spearman Correlation | 0.40 | 0.03* |
| | Clarity | Weighted Cohen's $\kappa$ | 0.13 | - |
| | | Spearman Correlation | 0.39 | 0.04* |
| | Relevance | Weighted Cohen's $\kappa$ | 0.34 | - |
| | | Spearman Correlation | 0.41 | 0.03* |
| | Depth of Argumentation | Weighted Cohen's $\kappa$ | 0.31 | - |
| | | Spearman Correlation | 0.50 | 0.01* |

## B.4 DISCUSSION

Among the evaluated models, largest judge, `Mistral-Large-Instruct-2407` showed the strongest alignment with human judgments, achieving moderate agreement across most criteria. Other models, `LLama3-70B-Instruct` and `Qwen2.5-72B-Instruct` exhibited lower alignment, with Kappa values indicating slight to fair agreement and weaker correlations overall.

We emphasize correlation over full alignment due to inherent subjectivity in both human and model assessments. When evaluating complex criteria like explanation quality, raters naturally vary in their interpretations and standards. Thus, achieving absolute agreement becomes not only challenging but potentially less meaningful as an alignment measure.

Correlation metrics better capture how LLMs and human raters rank explanations relative to each other. Strong correlations demonstrate that both groups identify similar patterns in quality, even if their absolute scores differ. This relative agreement is crucial for applications like ours that aim to differentiate between stronger and weaker outputs, rather than establish absolute quality benchmarks.

### B.5 LIMITATIONS

Despite providing valuable insights, our human evaluation has several limitations that should be acknowledged. Firstly, the sample size was relatively small (30 explanations from 10 questions sourced exclusively from the LogiQA dataset) which may limit the generalizability of our findings to other datasets or domains. Secondly, the inter-annotator reliability among human raters was only fair, with $\alpha$ values ranging from 0.16 to 0.47, indicating variability and inconsistency in human judgments. This variability highlights the third limitation: the inherent subjectivity in evaluating complex criteria such as explanation quality, logical coherence, and depth of argumentation. Different raters may apply diverse standards and interpretations, leading to discrepancies in assessments and underscoring the challenges in achieving high agreement in subjective evaluations.

## C AGREEMENT METRICS BETWEEN ALIGNMENT JUDGE AND EVALUATION JUDGE

To construct the preference dataset while ensuring a self-contained approach—where no external LLMs are involved in the evaluation or self-alignment process—the LLM undergoing self-alignment must act as the judge, ranking its own responses. These ranked responses are then utilized to create a self-preference dataset, which is subsequently used to perform a second alignment via DPO, yielding $\mathcal{M}_{\text{Rank}}$ and $\mathcal{M}_{\text{Anchor}}$. At this stage, only $\mathcal{M}_{\text{Base}}$ and $\mathcal{M}_{\text{SFT}}$ are available for comparison.

Although $\mathcal{M}_{\text{SFT}}$ demonstrates a decline in the quality of the explanations it generates compared to $\mathcal{M}_{\text{Base}}$, as measured through pairwise evaluations, this decline does not necessarily indicate a reduction in the model's ability to rank explanations. To evaluate this, we measured the agreement between the evaluation judge (`Llama3-70B-Instruct`) and the models $\mathcal{M}_{\text{Base}}$ and $\mathcal{M}_{\text{SFT}}$ using two metrics: Spearman Correlation and Weighted Cohen's $\kappa$. Results, presented in Table 4, show a noticeable decrease in agreement between $\mathcal{M}_{\text{SFT}}$ and the evaluation judge when compared to $\mathcal{M}_{\text{Base}}$ across all datasets.

Table 4: **Agreement Metrics Between Alignment Judge and Evaluation Judge.** This table compares the agreement between the evaluation judge (`Llama3-70B-Instruct`) and the models $\mathcal{M}_{\text{Base}}$ and $\mathcal{M}_{\text{SFT}}$ across various datasets. Two metrics were used: Spearman Correlation and Weighted Cohen's $\kappa$. A significant reduction in agreement is observed for $\mathcal{M}_{\text{SFT}}$ compared to $\mathcal{M}_{\text{Base}}$, particularly in Weighted Cohen's $\kappa$ values. Statistically significant correlations are marked with an asterisk (*).

| Dataset | Model | Metric | Value | p-value |
|---------|-------|--------|-------|---------|
| AQuA-Rat | $\mathcal{M}_{\text{Base}}$ | Weighted Cohen's $\kappa$ | 0.324 | - |
| | | Spearman Correlation | 0.353 | $0.001^*$ |
| | $\mathcal{M}_{\text{SFT}}$ | Weighted Cohen's $\kappa$ | 0.094 | - |
| | | Spearman Correlation | 0.079 | 0.475 |
| ARC-Challenge | $\mathcal{M}_{\text{Base}}$ | Weighted Cohen's $\kappa$ | 0.026 | - |
| | | Spearman Correlation | 0.378 | $2.3e^{-6*}$ |
| | $\mathcal{M}_{\text{SFT}}$ | Weighted Cohen's $\kappa$ | 0.017 | - |
| | | Spearman Correlation | 0.308 | $1.4e^{-4*}$ |
| LogiQA | $\mathcal{M}_{\text{Base}}$ | Weighted Cohen's $\kappa$ | 0.207 | - |
| | | Spearman Correlation | 0.404 | $0.001^*$ |
| | $\mathcal{M}_{\text{SFT}}$ | Weighted Cohen's $\kappa$ | 0.156 | - |
| | | Spearman Correlation | 0.312 | $0.014^*$ |
| OpenBookQA | $\mathcal{M}_{\text{Base}}$ | Weighted Cohen's $\kappa$ | 0.058 | - |
| | | Spearman Correlation | 0.302 | $2.1e^{-4*}$ |
| | $\mathcal{M}_{\text{SFT}}$ | Weighted Cohen's $\kappa$ | 0.048 | - |
| | | Spearman Correlation | 0.206 | $0.012^*$ |

# D  DATASET DETAILS

All datasets used in our experiments are established reasoning benchmarks. The questions, along with related context and answer options, were inserted into our template (provided in 11) and used as input prompts for the model.

## D.1  AQUA-RAT

`AQuA-Rat` (Ling et al., 2017) contains approximately 100,000 algebraic word problems, each accompanied by a natural language rationale explaining the solution steps. Each problem includes a question and five answer options. Due to computational constraints, we sampled 5,000 examples from the training set and used 254 test samples for our experiments. We utilized only the questions and answer options and excluded the provided rationales since our study aims to enhance the model's ability to generate self-explanations in natural language while performing the primary classification task, without relying on human-annotated reasoning patterns.

## D.2  ARC-CHALLENGE

The `ARC-Challenge` dataset (Clark et al., 2018) is a subset of the ARC dataset containing grade-school level, multiple-choice science questions. We used 1,119 training samples and 1,172 test samples from ARC-Challenge dataset. These samples are selected for being challenging, as they could not be answered correctly by either retrieval-based or word co-occurrence algorithms. In our experiments, we incorporated the questions and their corresponding four answer options. We omitted the associated corpus of sentences to focus purely on the model's reasoning capabilities rather than external knowledge retrieval.

### D.3 LOGIQA

`LogiQA` (Liu et al., 2020) consists of logical reasoning problems derived from the National Civil Servants Examination of China. The questions are designed to assess critical thinking and problem-solving abilities, requiring examinees to read a context passage and answer questions based on logical reasoning. In our experiments, we utilized English versions of 7,376 training samples and 651 test samples, including the context passages, questions, and answer options. The context passages were retained because they were integral to understanding and answering the questions.

### D.4 OPENBOOKQA

`OpenBookQA` (Mihaylov et al., 2018) features elementary-level science questions requiring multi-step reasoning and common knowledge application. The dataset simulates an "open book" exam setting to assess understanding beyond simple fact retrieval. It includes a corpus of scientific facts alongside multiple-choice questions, each with four answer options. In our experiments, we used 4,957 training samples and 500 test samples while excluding the related facts to align with our goal of developing self-explanation capabilities without leveraging pre-existing explanatory content.

## E INFERENCE PARAMETERS

Table 5 summarizes the inference parameters, including temperature and top-k, used for each component, such as the judge, consultant, and sampler.

Table 5: **Inference parameters per component**

| Component | Temperature | Top-k |
|-----------|-------------|-------|
| Judge | 0.0 | |
| Consultant | 0.5 | 0.9 |
| Sampler | 0.6 | 0.9 |

## F JUDGE COMPONENT

The judge model $\mathcal{M}_{\text{Judge}}$ evaluates the quality of self-explanation, denoted as $\varepsilon_i$, associated with an input prompt $x_i$. based on predefined criteria, which are elaborated in Section 2.1. The evaluation process proceed as follows:

1. For each criterion $\kappa$, $\mathcal{M}_{\text{Judge}}$ assigns a qualitative verdict $v_{i,\kappa}$ from the set {excellent, good, fair, poor, bad}. The prompt used by $\mathcal{M}_{\text{Judge}}$ is provided in Appendix F.2.

$$\mathcal{M}_{\text{Judge}}(x_i, \varepsilon_i) \to \{v_{i,\kappa}\} \text{ for } \kappa \in \{1, \ldots, K\}$$

2. Each verdict $v_{i,\kappa}$ is mapped to a numerical score $s_{i,\kappa}$ (see Appendix F.1).

3. The overall score for an explanation, $s_i$, is computed as the sum of scores across all criteria:

$$s_i = \sum_{k=1}^{K} s_{i,k}$$

### F.1 JUDGE SCORE MAPPING

Each verdict $v_{i,\kappa}$, assigned by $\mathcal{M}_{\text{Judge}}$ for criterion $\kappa$ on self-explanation $\varepsilon_i$ corresponding to prompt $x_i$, is mapped to a numerical score $s_{i,\kappa}$ as follows:

$$
s_{i,\kappa} = \begin{cases}
1.0 & \text{if } v_{i,\kappa} = \texttt{Excellent}, \\
0.8 & \text{if } v_{i,\kappa} = \texttt{Good}, \\
0.6 & \text{if } v_{i,\kappa} = \texttt{Fair}, \\
0.2 & \text{if } v_{i,\kappa} = \texttt{Poor}, \\
0.0 & \text{if } v_{i,\kappa} = \texttt{Bad}.
\end{cases}
$$

In this mapping, scores $s_{i,\kappa} \in \{1.0, 0.8, 0.6, 0.2, 0.0\}$ represent the numerical evaluation of the verdict, with higher values indicating better performance.

## F.2 JUDGE PROMPT

Table 6 presents the complete prompt that guides the judge model in evaluating the self-explanations.

Table 6: **Judge prompt**.

---

**Prompt Judge**

---

**System:** You are an expert judge tasked with evaluating the quality and correctness of a STATEMENT in response to a given QUESTION.

Your goal is to assess whether the STATEMENT accurately and comprehensively answers the QUESTION while providing sound reasoning and clear explanations. Be vigilant for any errors, misleading information, or gaps in the reasoning.

Evaluate the STATEMENT based on the following criteria:

- **Factual Accuracy:** Are all specific claims, data points, and facts in the STATEMENT verifiably correct?
- **Logical Coherence:** Does the STATEMENT follow a clear, logical sequence of reasoning without contradictions, and do all parts logically connect to form a cohesive and unified explanation?
- **Clarity and Comprehensibility:** Is the STATEMENT expressed clearly and precisely, using appropriate terminology and effectively explaining complex ideas?
- **Relevance and Completeness:** Does the STATEMENT fully address all relevant aspects of the QUESTION, ensuring that it directly responds to the specific context or requirements, leaving no key details omitted?
- **Depth of Argumentation:** Does the STATEMENT provide strong reasoning and credible evidence to support its conclusions?

For each criterion, select one of the following assessments:

- **EXCELLENT:** The STATEMENT exemplifies this criterion, setting a high standard.
- **GOOD:** The STATEMENT meets this criterion well, with only minor room for improvement.
- **FAIR:** The STATEMENT adequately meets this criterion but has noticeable areas for enhancement.
- **POOR:** The STATEMENT partially meets this criterion but requires significant revisions.
- **BAD:** The STATEMENT fails to meet this criterion.

Provide the assessments (EXCELLENT/GOOD/FAIR/POOR/BAD) for each criterion without additional justification. Respond with the following structure:

- **Factual Accuracy:** [Assessment]
- **Logical Coherence:** [Assessment]
- **Clarity:** [Assessment]
- **Relevance:** [Assessment]
- **Depth of Argumentation:** [Assessment]

---

**User:** Here is the question: **[QUESTION]**

Here is the statement you need to evaluate: **[STATEMENT]**

---

### F.3 CONSULTANT COMPONENT

In cases where the model $\mathcal{M}_{\text{SFT}}$ behaves consistently incorrectly for the input prompt $x_i$, we employ the model $\mathcal{M}_{\text{Base}}$ in a consultant role. Specifically, we provide the correct answer $y_i$ to the LLM and request an explanation $\varepsilon_i$ supporting this answer.

$$\mathcal{M}_{\text{Debater}}(x_i, y_i) \rightarrow \varepsilon_i$$

### F.4 CONSULTANT PROMPT

Table 7: **Consultant prompt**.

---

**Consultant prompt**

---

**System:** You will act as an expert consultant. You will be shown a QUESTION and one OPTION to argue for. Your goal is to construct convincing arguments supporting your option as correct.

Maintain a scholarly yet assertive tone, focusing on evidence-based analysis while crafting your arguments and answering questions. Avoid phrases like "In summary", "In conclusion", or similar at the end of your argument. Be as concise as possible.

Your argument must fulfill the following criteria:

- **Truthfulness:** Any factual information mentioned in your argument must be true and accurate.
- **Logical Consistency:** Your argument must be logically sound, free of logical fallacies or contradictions.
- **Clarity:** Your argument directly addresses and answers the question. It is clear, concrete, concise, and well-structured.

---

**User:** Here is the question: **[QUESTION]**

Here is option you need to back up: **[OPTION]**

---

## G PROMPTS INFERENCE PER TASK

Table 8: **Prompt for LogiQA task**.

---

**Prompt for LogiQA task**

---

**System:** You will be presented with a CONTEXT passage and a corresponding QUESTION with four answer CHOICES. Carefully read the passage to understand its content. Then, read the QUESTION and CHOICES thoroughly. Choose the correct CHOICE and explain your reasoning.

Your response will consist of two parts: an EXPLANATION followed by your selected CHOICE.

Enclose your explanation within tags as follows:
<explanation>[Your EXPLANATION here]</explanation>

Enclose your chosen choice (e.g., if the question has only 4 choices, then A, B, C, or D) within tags as follows:
<choice>[Your CHOICE here]</choice>

---

**User:** Context: **[CONTEXT]**

Question: **[QUESTION]**

Choices: **[CHOICES]**

---

## G.1 PROMPT FOR AQUA-RAT TASK

Table 9: **Prompt for AQuA-Rat task**.

---

**Prompt for AQuA-Rat task**

---

**System:** You will be given a QUESTION along with multiple answer CHOICES, involving a math problem that requires step-by-step reasoning to determine the correct answer. Carefully read the QUESTION and CHOICES. Choose the correct CHOICE and explain your reasoning.

Your response will consist of two parts: an EXPLANATION followed by your selected CHOICE.

Enclose your explanation within tags as follows:
<explanation>[Your EXPLANATION here]</explanation>

Enclose your chosen choice (e.g., if the question has only 4 choices, then A, B, C, or D) within tags as follows:
<choice>[Your CHOICE here]</choice>

---

**User:** Context: **[CONTEXT]**

Question: **[QUESTION]**

Choices: **[CHOICES]**

---

## G.2 PROMPT FOR ARC-CHALLENGE TASK

Table 10: **Prompt for ARC-Challenge task**.

---

**Prompt for ARC-Challenge task**

---

**System:** You will be presented a QUESTION with multiple answer CHOICES. Carefully read the QUESTION and CHOICES. Choose the correct CHOICE and explain your reasoning.

Your response will consist of two parts: an EXPLANATION followed by your selected CHOICE.

Enclose your explanation within tags as follows:
<explanation>[Your EXPLANATION here]</explanation>

Enclose your chosen choice (e.g., if the question has only 4 choices, then A, B, C, or D) within tags as follows:
<choice>[Your CHOICE here]</choice>

---

**User:** Context: **[CONTEXT]**

Question: **[QUESTION]**

Choices: **[CHOICES]**

---

### G.3 PROMPT FOR OPENBOOKQA TASK

Table 11: **Prompt for OpenbookQA task**.

---

**Prompt for OpenbookQA task**

---

**System:** You will be presented a QUESTION with multiple answer CHOICES. Carefully read the QUESTION and CHOICES. Choose the correct CHOICE and explain your reasoning.

Your response will consist of two parts: an EXPLANATION followed by your selected CHOICE.

Enclose your explanation within tags as follows:
<explanation>[Your EXPLANATION here]</explanation>

Enclose your chosen choice (e.g., if the question has only 4 choices, then A, B, C, or D) within tags as follows:
<choice>[Your CHOICE here]</choice>

---

**User:** Context: **[CONTEXT]**

Question: **[QUESTION]**

Choices: **[CHOICES]**

---

## H  GENERATED INSTRUCTIONS

Table 12: **Distribution of anchor categories:** This table presents the distribution of the categories—Consistently Correct (CC), Consistently Incorrect (CI), and Variable (V)—across datasets used during the DPO alignment phase of $\mathcal{M}_{\text{Anchor}}$ .

| Dataset | Category | Samples | Ratio (%) |
|---|---|---|---|
| AQuA-Rat | V | 1196 | 41.17 |
| | CC | 1010 | 34.77 |
| | CI | 699 | 24.06 |
| ARC-Challenge | V | 62 | 8.09 |
| | CC | 645 | 84.20 |
| | CI | 59 | 7.70 |
| LogiQA | V | 1251 | 26.86 |
| | CC | 2487 | 53.39 |
| | CI | 920 | 19.75 |
| OpenbookQA | V | 176 | 5.13 |
| | CC | 3178 | 92.60 |
| | CI | 78 | 2.27 |

