# OpenReview forum: "Anchored Alignment for Self-Explanations Enhancement"
_ICLR.cc/2025/Conference — Submitted to ICLR 2025_

### Official Review · Reviewer_7CFp · 2024-11-01

**Soundness:** 4
**Presentation:** 4
**Contribution:** 3
**Rating:** 8
**Confidence:** 4

**Summary:**

The authors propose a pipeline to generate self-explanation without gold explanations. The process is divided into three main steps: i) explanation evaluation, ii) creating a self-instruct dataset, and iii) aligning the model with Anchor Preference Pairs which are "better" pairs that are refined by categorizing the model's output in three different categories and are then utilized in DPO for model alignment. The main achievement of this research is mitigating the effect of fine-tuning which causes degradation in explanation quality without harming the task accuracy.

**Strengths:**

- The paper targets a common yet important problem in the field of explanations.
- A comprehensive evaluation of the explanations through the introduced metrics.
- Using LLM as a judge to evaluate self-explanations w.r.t. the given metrics.

**Weaknesses:**

N/A

**Questions:**

- How do you choose between two or more explanations with the same score when calling min/max to generate Preference Pairs? My question is regarding line 225.

---

> ### Author Response · Authors · 2024-12-01
>
> We sincerely appreciate your insightful comments and constructive feedback on our work.
>
> ---
> **Q1**: How do you choose between two or more explanations with the same score when calling min/max to generate Preference Pairs? My question is regarding line 225.
>
> **Answer Q1:**  It may vary depending on the strategy used: Consistently Correct (CC), Variable (V), or Consistently Incorrect (CI). However, in the specific case on line 225 (CC strategy), explanations with the same highest score will be assigned to the set
> Aw, and the rest will be assigned to Al. At the end of the algorithm, we sample a winning and losing explanation from these two sets.

---

### Official Review · Reviewer_5ZbZ · 2024-11-04

**Soundness:** 2
**Presentation:** 2
**Contribution:** 2
**Rating:** 3
**Confidence:** 5

**Summary:**

This paper introduces Alignment with Anchor Preference Pairs, a method to improve an LLM's reasoning capability when there is no CoT rationale available. The key idea is to employ the LM as both a judge and response generator. By dividing into three cases, the authors empirically show that their method is effective compared to naively constructing the preference dataset based on the judge model's prediction

**Strengths:**

* For employing LLM-as-a-Judge, the authors use 5 evaluation criteria and include an ablation experiment in Figure 1 to compare which one was effective. This is very insightful.

* The methodology is clear and very intuitive.

**Weaknesses:**

* If the main point is exploring how to utilize CoT rationales when they are not available, I think there should also be a upper-bound score, which is curating CoT rationales from a stronger teacher model (for instance, Llama-3.1-70B-Instruct, which was used as the evaluator). Without that, it is hard to tell how effective the method is compared to a strong baseline which is distillation.

* There isn't enough ablation experiments to support the design choices in Section 3.3. For example:
   * When M_sft consistently produces correct answers, preference pairs are constructed based on the quality of the explanations. What if we remove these instances?
   * When there are a mix of correct and incorrect predictions, the explanations with the highest scores assigned by the judge model is employed. What are the trends like if counterfactually choosing instances where the judge model provides slightly lower scores?

* The evaluation is done with pairwise ranking (O(N^2)), making it computationally expensive to employ in diverse settings.

* The explanation in line 355 - 358 regarding how M_anchor could outperform M_rank lacks insight or details. Could you further elaborate on this?

* The algorithm still requires ground-truth annotations of the datasets (even if it doesn't require a CoT rationale).

**Questions:**

* In Section 3.2, why did you use the name "Self-Instruction Creation"? Aren't the inputs fixed? Perhaps it should be called "Self-Preference Data Creation" instead?

* I'm not sure how the method differs a lot from the "Self-rewarding LM" paper. Could you elaborate on this?

* Please see the questions in weaknesses

---

> ### Author Response · Authors · 2024-12-02
>
> We sincerely appreciate your insightful comments and constructive feedback on our work. Your suggestions have been invaluable in helping us identify areas for improvement and better articulate the contributions of our paper.
>
> ---
> **Addressing W1:**  We agree with the reviewer that distillation can serve as a strong baseline, especially in scenarios where annotated rationales are unavailable. However, distillation does not satisfy our design constraint that the approach must be self-contained—specifically, no stronger models should be used for alignment, only for evaluation. As such, we believe using distillation would not constitute a fair baseline.
>
> In contrast, selecting pairs for the preference dataset to obtain M_{Rank}—our chosen baseline—is based solely on scores, without relying on stronger models. This approach aligns with how state-of-the-art self-alignment methods are implemented in the literature.
>
> ---
> **Addressing W2:**  To better support the design choices outlined in Section 3.3, we followed the reviewer’s recommendation and added an ablation study to examine the effects of the strategies used to build the self-preference dataset. In Table 1, we present results for three versions of
> M_{Anchor}: CC, CC+V, and CC+V+CI.
>
> Additionally, we expanded the evaluation of self-explanation, originally conducted only with Llama3-70B-Instruct, to include Mistral-Large-Instruct-2407 and Qwen2.5-72B-Instruct. This extension further demonstrates the robustness of our approach.
>
> ---
> **Addressing W3:**  We respectfully disagree with this point. The computationally intensive part of the process occurs during inference, where the judge assigns scores to each explanation—a step with O(N) complexity. Once the scores for all explanations are computed, the pairwise evaluation (O(N^2)) is performed, but this step is not computationally demanding.
>
> The computational expense would be significant if the judge were to directly rank pairs of explanations (e.g., determining which is better, A or B), as this would require O(N^2)mcomparisons during inference.
>
> ---
> **Addressing W4:**
>
> The (CI) and (V) strategies ensure that, assuming the self-explanation is faithful, the winning explanation aligns with the ground-truth label. For the (V) strategy, this is achieved by sampling the winning explanation from the set Bw, which contains explanations associated with correct predictions. In the case of the (CI) strategy, M_{Debater} is used to provide arguments that explicitly support the ground-truth label for the winning explanation.
>
> In contrast, M_{Rank} selects pairs for the preference dataset solely based on scores assigned by judges during its creation. As a result, it does not guarantee that the winning explanation will align with the ground-truth label. The likelihood of selecting cases in the preference dataset where the winning explanation does not match the ground-truth label increases, particularly when M_{SFT} exhibits consistently incorrect or variable behavior among the given inputs.
>
> In this sense, we can say that the winning explanations in the Anchor preference dataset are more truthful compared to the winning explanations in the Rank preference dataset. We believe this is why M_{Anchor} performs better than M_{Rank} in cases where V and CI cases are abundant (i.e., when lambda is high, where lambda = 100%*(V + CI)/(CC + V + CI)).
>
> This explanation has been added in Section 4.1.4.
>
> ---
> **Addressing W5:**  It is a common challenge in real-world applications that datasets containing explanatory rationales are often scarce or prohibitively expensive compared to classification datasets. In such scenarios, classification datasets can be leveraged for domain-specific adaptation to a particular task (the primary task). However, we found that supervised fine-tuning (SFT) can negatively impact the quality of explanations—secondary tasks. Therefore, our goal was to enhance the quality of explanations (which were affected during the SFT phase) while preserving the accuracy gains achieved during that phase.
>
> Since the dataset used for domain adaptation already contains information about the correct choice (acting as a probe), we aimed to leverage this probe to improve the creation of the self-preference dataset (through the proposed Anchor alignment). While our approach does require ground-truth annotations from the classification task, such labels are inherently available in the domain adaptation scenarios described.
>
> ---
> **Answer Q1:** Thanks to the reviewer for pointing out the issue. It has been corrected and replaced with 'Self-Preference Data Creation.'
>
> ---
> **Answer Q2:** Since the dataset used for domain adaptation already contains information about the correct choice (acting as a probe), we aimed to leverage this probe to improve the creation of the self-preference dataset. The (CI) and (V) strategies utilize this information to ensure that the winning explanation aligns with the ground-truth label (see Section 4.1.4)

---

### Official Review · Reviewer_VaYZ · 2024-11-04

**Soundness:** 3
**Presentation:** 4
**Contribution:** 3
**Rating:** 6
**Confidence:** 4

**Summary:**

The paper presents a framework for evaluating the quality of self-explanations from language models using criteria like coherence, clarity, and accuracy, with an LLM acting as the judge. It introduces the "Alignment with Anchor Points" method to create reliable preference pairs and uses an alignment strategy that boosts performance while maintaining explanation quality, without needing human annotations.

**Strengths:**

**S1**. The paper presents a framework for assessing the quality of self-explanations based on various criteria, including logical coherence, clarity, depth of argumentation, and factual accuracy, using an LLM as the evaluator.

**S2**. The authors provide an analysis of the relationship between alignment methods and self-explanation quality, based on their evaluation framework.

**S3**. They introduce the "Alignment with Anchor Points" method to create high-quality preference pairs by grouping them according to the consistency of their accuracy, along with an end-to-end alignment strategy that enhances downstream performance while preserving self-explanation quality, without requiring human rationale annotations.

**S4**. The paper is well-written and easy to read.

**Weaknesses:**

**W1**. Although the authors state in Line 69 that their approach differs from faithfulness, they should be more precise when articulating the aim of their approach. The phrase "effectively conveying the model's reasoning" is broad and can be interpreted as faithfulness. A more suitable term might be "improving the plausibility of explanations," as this study aims to replace human-annotated preference pairs with LLM annotations, and the criteria selected resemble those that make explanations plausible to humans. An explicit discussion of where these criteria fit within the distinction between faithfulness and plausibility would be beneficial [1]. Additionally, while this alignment scheme might incidentally improve the faithfulness of explanations, it is not necessarily required to do so. It would be interesting to examine the correlation between each criterion dimension and faithfulness. References [2, 3] could be used for evaluating the faithfulness of explanations.

**W2**. The paper lacks details about the dataset used. A section, either in the main text or the appendix, should briefly summarize the tasks involved.

**W3**. Since the study relies solely on one base model, LLama-3-8B-Instruct, it is unclear if the proposed method's improvements are consistent across different model families or sizes. For example, the marginal improvement may be reduced in larger models.

**W4**. As mentioned in the limitations, using the base model as the judge during the creation of the self-instruct dataset may fail to capture certain nuances as the model evolves. However, the authors did not provide evidence to show whether this is a practical concern. An analysis comparing how often the M_{base}, M_{Anchor} and the evaluation judge agree on evaluation scores would be helpful.

**W5**. The paper lacks an ablation study that uses a different LLM as the judge for evaluation.

**W6**. The method is limited to classification tasks, as mentioned in the limitations section. However, it could potentially be extended to generation tasks using a similar approach, where outputs are grouped based on their similarity to ground-truth answers. Although this may be more complex than for classification tasks, discussing such possibilities could strengthen the paper.

**W7**. Figure 1 should not use radar charts, as they are difficult to interpret and can be misleading. Replacing each radar chart with a bar plot would make the data much easier to interpret.

**References**

1.  Jacovi, Alon and Yoav Goldberg. “Towards Faithfully Interpretable NLP Systems: How Should We Define and Evaluate Faithfulness?”

2. Lanham, Tamera et al. "Measuring Faithfulness in Chain-of-Thought Reasoning"

3. Parcalabescu, Letitia and Anette Frank. “On Measuring Faithfulness or Self-consistency of Natural Language Explanations.”

**Questions:**

**Q1**. Why does SFT result in a degradation in explanation quality?

**Q2**. In Figure 2, Section 4.2.4, how is $\lambda$ calculated? Is it  the raw number of preference pairs selected under the consistently-incorrect or variable strategies? How do you scale $\lambda$? Does the total dataset size increase as $\lambda$ increases, or is it kept constant? Could the relative improvement be solely due to a decrease in by decrease in M_{Rank} performance?

---

> ### Author Response · Authors · 2024-12-01
>
> We sincerely appreciate your insightful comments and constructive feedback on our work. Your suggestions have been invaluable in helping us identify areas for improvement and better articulate the contributions of our paper. We have tried to address most of the highlighted weaknesses and worked to strengthen the paper accordingly.
>
> ---
> **Addressing W1:** We greatly appreciate the reviewer’s insightful observation and have incorporated the proposed changes with some modifications. While an explanation may be considered plausible if it aligns with human reasoning, we believe this notion does not fully capture the objective of our approach. Simply resembling human reasoning does not necessarily make an explanation good.
>
> To address this, we introduce the term holistic explanation ( Section 2.1.), which we define as one that excels across multiple criteria: Logical Coherence, Clarity, Relevance, Depth of Argumentation, and Factuality. Together, these criteria collectively determine what constitutes a high-quality explanation.
>
> Furthermore, we have included the reviewer’s recommendation to explicitly discuss how these criteria fit within the distinction between faithfulness and holistic explanation, thereby providing greater clarity on their role in our framework.
>
> ---
> **Addressing W2:** Following the reviewer’s recommendation, details about the dataset used in the paper have been added to Appendix D.
>
> ---
> **Addressing W3:** We acknowledge the relevance of this weakness; however, due to time and computational constraints, we are unable to address it in the current work. Nonetheless, we will carry this point forward for consideration in our future research efforts.
>
> ---
> **Addressing W4:** To construct the preference dataset while maintaining a self-contained approach, the LLM undergoing self-alignment must serve as the judge, ranking its own responses. At this stage, only M_{Base} and M_{SFT} are available for comparison.
>
> Using the base model as the judge during the creation of the self-preference dataset addresses the issue that M_{SFT}  demonstrates a decline in explanation quality (see Section 4.1.1) and reduced agreement with the evaluation judge (Appendix C). In response to the reviewer’s suggestion, we have added an agreement test in Appendix C, comparing M_{Base} and M_{SFT} with respect to the evaluation judges. We focused on these models for the test instead of M_{Anchor}  because they are the models available to rank the explanations that will be used to create the preference dataset (considering a single iteration, as outlined in the paper). These results provide evidence for why we selected M_{Base} as the judge.
>
> However, in scenarios involving multiple iterations of alignment, the use of M_{Base} to rank explanations during the second or third iterations of preference dataset creation would need to be reevaluated, as the model’s performance and alignment may evolve over time.
>
> ---
> **Addressing W5:** We expand the evaluation of self-explanation, which was initially conducted only with Llama3-70B-Instruct, to include Mistral-Large-Instruct-2407 and Qwen2.5-72B-Instruct. This extension helps to better demonstrate the robustness of our approach (see Table 1).
>
> Additionally, we present an ablation study on the effects of dataset composition. In Table 1, we report results for three versions of
> M_{Anchor}: CC, CC+V, and CC+V+CI.
>
> ---
> **Addressing W6:** We appreciate the reviewer’s suggestion. While we did consider this extension, we chose not to include it in the current version of the paper, as we were concerned that other reviewers might prefer avoiding claims that are not empirically supported. However, we agree that exploring this possibility could be valuable, and we will consider it in future work, where we can provide the necessary empirical evidence to support such claims.
>
> ---
> **Addressing W7:**
> We appreciate the reviewer’s suggestion. While we initially thought replacing the radar charts with bar plots would be straightforward, we found that displaying 5 criteria across 4 models and 4 datasets in bar plots would require too much space and exceed the page limit. Therefore, we were forced to retain the radar charts.
>
> ---
> **Answer Q1:** Building on evidence that supervised fine-tuning can improve performance on specific tasks at the expense of a model's generalization abilities [1], we hypothesize that this decline occurs because the task of selecting predefined answers does not inherently encourage the model to articulate its reasoning, leading to a specialization that diminishes the quality of the generated explanations.
>
> [1]. Unveiling the Generalization Power of Fine-Tuned LLMs  (Yang et al., NAACL 2024)
>
> ---
> **Answer Q2:**
> - Lambda = 100*(CI+ V) / (CC + V +CI) (%).
> - Lambda represents the composition of strategies in the Anchor dataset (We have extended the explanation of the use of lambda in Section 4.1.4.)
> - The Rank and Anchor datasets are comparable in size.

---

### Official Review · Reviewer_vG9v · 2024-11-06

**Soundness:** 3
**Presentation:** 3
**Contribution:** 2
**Rating:** 5
**Confidence:** 3

**Summary:**

This paper introduces a new self-explanation alignment method, called Alignment with Anchor Preference Pairs, to improve both exact answer prediction and explanation quality in response to given questions. The main idea is to categorize preference pairs into three types: consistently correct prompts, consistently incorrect prompts, and a mixed category. The authors also propose a new evaluation framework for explanations, measuring five aspects - logical coherence, clarity, relevance, depth of argumentation, and factual accuracy - using other large language models (LLMs). Experiments on four different datasets show that the proposed method outperforms existing self-alignment techniques in generating better explanations, as assessed by the new evaluation framework. Although answer accuracy was somewhat higher than baseline methods, the difference was not statistically significant. The authors further analyze the effect of preference pair category distribution and find that selecting preference pairs from consistently incorrect prompts and the mixed set improves performance over baselines.

**Strengths:**

The main strength of this paper lies in its introduction of a novel and well-executed modeling of self-explanation alignment. The concept is both sound and straightforward, making it applicable to other QA tasks and LLM models. In terms of clarity, the paper is generally well-written, though certain aspects could benefit from further elaboration.

**Weaknesses:**

One main concern is the effectiveness of the proposed method in answer accuracy and explanation quality. While M_{Anchor} outperforms M_{Rank}, the differences shown in Table 1 are not statistically significant. Are the reported ± values standard deviations or standard errors? Regarding explanation quality, it would strengthen the paper to validate the new evaluation framework by measuring its correlation with human judgments. Even if LLMs perform similarly to human raters, the reliability of a new evaluation metric needs to be confirmed.

**Questions:**

- It would be helpful to test the proposed method with one or two more LLMs to demonstrate its robustness and applicability.
- Can the lambda in line 423 be optimized as a hyperparameter? In the current setup, lambda is chosen from the dataset, but controlling lambda as a hyperparameter (by adding a condition after line 18 in Algorithm 1) might improve model performance, especially given Figure 2’s positive correlation between lambda and model performance.
- Why does the performance of M_{Anchor} vary across the four datasets? While lambda partially explains these variations, understanding other influencing factors could reveal ways to further improve performance.
- Are the symbols “s” in lines 113 and 193 referring to the same concept?

---

> ### Author Response · Authors · 2024-12-01
>
> We sincerely appreciate your insightful comments and constructive feedback on our work. Your suggestions have been invaluable in helping us identify areas for improvement and better articulate the contributions of our paper. We have addressed most of the highlighted weaknesses and worked to strengthen the paper accordingly.
>
> ---
> **Weakness 1:** One main concern is the effectiveness of the proposed method in answer accuracy and explanation quality. While M_{Anchor} outperforms M_{Rank}, the differences shown in Table 1 are not statistically significant. Are the reported ± values standard deviations or standard errors?
>
> **Addressing Weakness 1:**
> The self-preference dataset used during the DPO phase is specifically designed to improve explanation quality rather than maximize accuracy. However, it is crucial to ensure that enhancing the model's self-explanation capabilities does not negatively impact its primary task performance, as measured by classification accuracy. To evaluate this, we compute the average classification accuracy and standard deviation across multiple evaluation runs (N=16), as shown in Table 1. In the table, the highest value is bolded, and results marked with (*) indicate no statistical difference from the top performer.
>
> In essence, our goal was to enhance the quality of explanations (affected during the SFT phase) while preserving the accuracy gains achieved during that phase.
>
> Interestingly, we observe a significant boost in accuracy for the AQuA Rat dataset, which suggests a dependency between performance and task type. In Section 4.1.4, we propose a plausible explanation for the source of this accuracy improvement in this specific scenario.
>
> These clarifications were added in Sections 4.1.2 and 4.1.4.
>
> ---
> **Weakness 2:** Regarding explanation quality, it would strengthen the paper to validate the new evaluation framework by measuring its correlation with human judgments. Even if LLMs perform similarly to human raters, the reliability of a new evaluation metric needs to be confirmed.
>
> **Addressing Weakness 2:**  To address this point, we conducted a study in Appendix B to evaluate the correlation between human judgment and LLM evaluations across the criteria proposed in our framework: Factual Accuracy, Logical Coherence, Clarity, Relevance, and Depth of Argumentation. We sampled 30 explanations generated by either aligned models or the base model, using 10 distinct questions sourced from the LogiQA dataset. LogiQA was chosen because it exhibited variability across all dimensions.
>
> We compared human judgment (3 human raters) against three LLMs acting as evaluators: Mistral-Large-Instruct-2407, Qwen2.5-72B-Instruct, and Llama3-70B-Instruct. Spearman correlation analysis revealed moderate, positive, and significant correlations across all three models (see Table 3) for each evaluation criterion (except for one case, which was not significant). Additionally, we included  Weighted Cohen’s Kappa coefficient to measure agreement between humans and LLMs, which showed slight to moderate agreement.
>
> We emphasize that for evaluation purposes, when comparing the quality scores of explanations, relative agreement between human and LLM judges is sufficient. In this context, Spearman's rank correlation is a more appropriate metric for assessing this alignment.
>
> ---
>
> **Q1:** It would be helpful to test the proposed method with one or two more LLMs to demonstrate its robustness and applicability.
>
> We extend the evaluation of self-explanation, initially performed only by Llama3-70B-Instruct, to include Mistral-Large-Instruct-2407 and Qwen2.5-72B-Instruct, in order to better demonstrate the robustness of the approach (see Table 1).
>
> ---
>
> **Q2:** Can the lambda in line 423 be optimized as a hyperparameter?
>
> We define lambda as the proportion of the self-preference dataset used to align M_{Anchor}, which corresponds to preference pairs selected under the (CI) or (V) strategies (lambda=100%*(CI+V)/(CC+CI+V)). This metric, lambda, provides insight into how the composition of the self-preference dataset for M_{Anchor} differs from that of M_{Rank} (Baseline). Understanding lambda helps explain why the performance improvement of M_{Anchor} is task-dependent (see section 4.1.4).
>
> We present an ablation study on the effects of dataset composition. In Table 1, we provide results for three versions of M_{Anchor} (CC, CC+V, and CC+V+CI). Our findings show that the CC+V+CI strategy consistently performs the best. Therefore, we conclude that treating lambda as a hyperparameter will not provide additional benefits.
>
> ---
> **Q3:** Are the symbols “s” in lines 113 and 193 referring to the same concept?
>
> Yes, it refers to the scores of an explanation. Nevertheless, it is important to clarify that scoring is conducted in two stages: during the creation of the self-preference dataset and during the evaluation process.

---

### Author Response · Authors · 2024-12-04

Dear Reviewers,

We sincerely thank you for your enthusiastic and supportive reviews of our paper, as well as the insightful comments and questions that have guided us in improving the clarity and technical details of our work. We have carefully considered your feedback and prepared a point-by-point response to each comment. We have addressed the major weaknesses highlighted in the reviews to strengthen the paper. We hope that these improvements will encourage you to reconsider and increase your scores.

---

### Meta-Review · Area_Chair_YgFV · 2024-12-21

**Metareview:**

This paper presents a method for improving methods' self-explanation abilities. It produces sets of preference pairs for alignment to find data that will improve these abilities. This is done by separating the examples into several cases (consistently correct, consistently incorrect, and variable) and using a different strategy for each. The strategy for incorrect prompts resembles how STaR (Zelikman et al.) handles incorrect predictions. The paper evaluates explanations with LLM-as-a-judge across several factors. Improvements are shown on mathematical, scientific, and logical reasoning datasets.

This paper explores an interesting and timely problem. The proposed division of examples into the different cases is a nice way to handle it. However, I think there is some circularity in the work: an LLM judge is used in the process of constructing the preference pairs, and then again for evaluation. Although it is the M_Base model, I think this is still circular. Because of shared training data, even different LLMs (especially Llama 3 8B and Llama 3 70B) will be correlated, and I don't have high confidence that the reported gains represent robust improvements in explanation quality. I think more qualitative and human analysis is needed to show that the improvements in explanations represent something real as opposed to surface-level stylistic factors, which LLM judges have been shown to be biased by (e.g., https://lmsys.org/blog/2024-08-28-style-control/ ).

Appendix B makes an argument about correlation with human judgments. The correlations are present, but weak. It is possible that improvements in the judge scores may not be reflected by human evaluation, particularly since the judges are used in the process.

**Additional Comments On Reviewer Discussion:**

The response effectively addressed questions about the gain in accuracy vs. explanation quality (vG9v) and the overall goal of the work (VaYZ).

As I said, I am not quite convinced by the correlation analysis in Appendix B (in the response to vG9v). In my view, this is one of the biggest weaknesses of the paper.

Most other minor points are convincingly addressed, with the addition of some new ablation studies.

---

### Decision · Program_Chairs · 2025-01-22

Reject